# Remote and Proximal Sensors Data Fusion: Digital Twins in Irrigation Management Zoning

**DOI:** 10.3390/s24175742

**Published:** 2024-09-04

**Authors:** Hugo Rodrigues, Marcos B. Ceddia, Wagner Tassinari, Gustavo M. Vasques, Ziany N. Brandão, João P. S. Morais, Ronaldo P. Oliveira, Matheus L. Neves, Sílvio R. L. Tavares

**Affiliations:** 1Institute of Food and Agricultural Sciences, University of Florida, McCarty Hall, 1604 McCarty Dr 1008, Gainesville, FL 32603, USA; 2AgroTechnologies and Sustainability Department, Federal Rural University of Rio de Janeiro, BR-465, Km 7, Seropédica 23897-000, RJ, Brazil; marcosceddia@gmail.com; 3Department of Mathematics, Federal Rural University of Rio de Janeiro, Rio de Janeiro 22460-000, RJ, Brazil; wtassinari@gmail.com; 4Brazilian Agricultural Research Company (Embrapa) Soils, Rio de Janeiro 22460-000, RJ, Brazil; gustavo.vasques@embrapa.br (G.M.V.); ronaldo.oliveira@embrapa.br (R.P.O.); silvio.tavares@embrapa.br (S.R.L.T.); 5Brazilian Agricultural Research Company (Embrapa) Cotton, Rua Osvaldo Cruz, 1143, Centenário 58428-095, CG, Brazil; ziany.brandao@embrapa.br (Z.N.B.); joao.morais@embrapa.br (J.P.S.M.); 6Postgraduate Program in Agronomy—Soil Science, Federal Rural University of Rio de Janeiro, BR-465, Km 7, Seropédica 23897-000, RJ, Brazil; matheul.dasneves@gmail.com

**Keywords:** spatial statistics, geographically weighted regression, pedometry, digital soil mapping

## Abstract

The scientific field of precision agriculture employs increasingly innovative techniques to optimize inputs, maximize profitability, and reduce environmental impact. However, obtaining a high number of soil samples is challenging in order to make precision agriculture viable. There is a trade-off between the amount of data needed and the time and resources spent to obtain these data compared to the accuracy of the maps produced with more or fewer points. In the present study, the research was based on an exhaustive dataset of apparent electrical conductivity (aEC) containing 3906 points distributed along 26 transects with spacing between each of up to 40 m, measured by the proximal soil sensor EM38-MK2, for a grain-producing area of 72 ha in São Paulo, Brazil. A second sparse dataset was simulated, showing only four transects with a 400 m distance and, in the end, only 162 aEC points. The aEC map via ordinary kriging (OK) from the grid with 26 transects was considered the reference, and two other mapping approaches were used to map aEC via sparse grid: kriging with external drift (KED) and geographically weighted regression (GWR). These last two methods allow the increment of auxiliary variables, such as those obtained by remote sensors that present spatial resolution compatible with the pivot scale, such as data from the Landsat-8, Aster, and Sentinel-2 satellites, as well as ten terrain covariates derived from the Alos Palsar digital elevation model. The KED method, when used with the sparse dataset, showed a relatively good fit to the aEC data (R^2^ = 0.78), with moderate prediction accuracy (MAE = 1.26, RMSE = 1.62) and reasonable predictability (RPD = 1.76), outperforming the GWR method, which had the weakest performance (R^2^ = 0.57, MAE = 1.78, RMSE = 2.30, RPD = 0.81). The reference aEC map using the exhaustive dataset and OK showed the highest accuracy with an R^2^ of 0.97, no systematic bias (ME = 0), and excellent precision (RMSE = 0.56, RPD = 5.86). Management zones (MZs) derived from these maps were validated using soil texture data from clay samples measured at 0–10 cm depth in a grid of 72 points. The KED method demonstrated the highest potential for accurately defining MZs for irrigation, producing a map that closely resembled the reference MZ map, thereby providing reliable guidance for irrigation management.

## 1. Introduction

The main scope of precision agriculture methods is to reduce the inputs necessary for planting by identifying the heterogeneities of the factors controlling productivity [1]. Much has been studied in precision agriculture to locate more or fewer productive regions [2,3] to reduce productivity costs and maximize profit. Some tools have been implemented to identify these regions with characteristics prone to generating adequate productivity conditions for the desired crop.

According to the U.S. Soil Survey and Classification manual, proximal soil sensing refers to equipment sensitive to some soil property. Based on mathematical modeling, it represents soil properties that are generally difficult to measure [4].

Before these tools became recognized as a method of efficiently identifying the patterns of the soils, some researchers wrote extensively about the methods of obtaining data via proximal sensors of the conductivity meter type [5,6,7,8,9,10,11,12,13,14]. The results described by them served as a subsidy and support for the scientific field that is currently known as precision agriculture. The potential of using non-contact sensors with the soil to recognize its variabilities through electromagnetic properties, gamma spectrometry, or even with other radiation ranges, such as visible and infrared variations, has been extensively reported [15,16,17,18,19,20,21].

Allied to proximal sensing methods, remote sensing, in essence, involves a similar piece of equipment to proximal sensing. However, it consists of a sensor embedded in a satellite with another potential range and sensitivity that obtains data with a higher spatial resolution [19,22,23,24]. When the data from remote and proximal sensors are combined, the data fusion methodology is defined [25,26,27,28,29,30,31,32].

In precision agriculture, another demand is obtaining soil data to analyze with the proximal and remote sensors data. In this sense, obtaining other data sources does not dispense with obtaining soil samples, since the relationships between the measured properties must be established [21,33].

Imagine the scenario of a farm with 1000 rainfed hectares and five irrigation pivots covering 500 ha each. The irrigation pivots and the areas without this equipment have slightly distinct geomorphologies with slope variation within the range of 3% slope. Considering the factors of soil formation as being a combination of relief, climate, organisms, parental material, existing soil types, and anthropic effects acting together in a given time in the scenario of precision agriculture, the most significant factors for understanding the dynamics between the properties measured by remote and proximal sensors and the soil properties that reflect the crucial parameters for irrigation control will be affected mainly by the type of soil [34,35,36].

In this scenario, the person responsible for mapping this critical information can combine the data from remote sensors with proximal sensors to improve the maps of soil properties such as texture, density, porosity, depth, etc. The person in charge must collect the proximal sensing data using equipment attached to the vehicle, such as conductivity and non-contact susceptibility meters, commonly available to consumers [37,38].

From then on, the person responsible must decide answers to the typical questions that still do not exist in the literature: What is the way to drive the vehicle to collect those data? How much data should I collect? What is the ideal spacing between my curves? The person in charge must consider the trade-off to obtain data ranging from exaggeration regarding the time and fuel spent to sub-represented soil properties data from spacing too far and collecting too few points.

In this Scopus, the hypothesis for this article arises: Is it reliable to reduce the amount of data collected using a proximal conductivimeter by combining its data with remote sensing data? To prepare an answer to this hypothesis, a simulation of different driving routes to obtain the proximal sensor data can be analyzed as a digital twin [39,40].

Therefore, we intend to answer the hypothesis by obtaining apparent electrical conductivity data from a commercial electromagnetic sensor commonly used in precision agriculture, such as the EM38-MK2.

We collected sensor data considering the most detailed sampling design possible, covering 72 ha of bean plantation in São Paulo, Brazil, totaling 25 transects. A second grid was simulated by maintaining only four transects of apparent electrical conductivity data to represent the scenario of obtaining proximal sensor data to configure a sparse grid.

To answer the hypothesis, the objectives of the article are (a) to map the apparent electrical conductivity data using the ordinary kriging method from an exhaustive grid with 25 lines and consider it a reference; (b) to map the apparent electrical conductivity data using kriging with external drift in association with remote sensing data; (c) to map the apparent electrical conductivity data using geographically weighted regression in association with remote sensing data; (d) to evaluate the accuracy of the maps produced in (a–c); (e) to define irrigation management zones based on clustering using the maps produced in (a–c); and (f) to evaluate the potential for distinguishing areas of management for irrigation from the maps produced in (a–c) using texture data, such as clay concentration.

## 2. Materials and Methods

### 2.1. Brief Presentation of the Methodology

The flowchart in Figure 1 illustrates the two parallel methodologies employed in the study, providing a clear and comprehensive overview of the Digital Twins approach applied to original and simulated aEC datasets.

The original aEC dataset undergoes a straightforward mapping process using the ordinary kriging (OK) method. A k-means clustering algorithm generates a management zone map, dividing the area into three distinct zones. The efficiency of these zones is then assessed by comparing them with soil sampling locations, specifically evaluating clay content.

In contrast, the simulated aEC dataset, representing a more data-sparse scenario, follows a more complex methodology. Initially, a preselection of covariates begins with Pearson’s correlation to identify covariates with the highest correlation to aEC. A stepwise selection process is then employed to reduce multicollinearity, ultimately selecting the covariate set with the lowest Bayesian information criterion (BIC) for further analysis. This refined set of covariates is then used in three different mapping methods: Ordinary kriging (OK), kriging with external drift (KED), and geographically weighted regression (GWR). Each method produces separate maps, which are then used to create three k-means maps, each defining three management zones. The efficiency of these zones is also evaluated by comparing them with soil sampling locations based on clay content, like the original approach.

### 2.2. Study Area

The area of the irrigation pivot is 72 ha and is in the Zacharias watershed, in the municipality of Itaí, state of São Paulo, Brazil, with central coordinates 23.5854° S and 48.9395° W, with an elevation of approximately 685 m altitude (Figure 2). According to Köppen-Geiger, the region’s climate presents the Aw weather pattern. The average annual rainfall for Itaí—SP is 119 mm, and the average annual temperatures range from 16 to 26 °C [41].

The soil types were described explicitly as Ferralsols, with the texture in the clayey surface layer (515 g kg^−1^) and very clayey (600 g kg^−1^) in the subsurface. Usually, the study area is planted with minimal soil disturbance, being classified as a no-tillage area on straw. The crop rotation method is implemented to improve the physical structure of the soil aggregates and to improve the incorporation of nutrients that have been removed by leaching erosion or used by the plants in the last harvest [42]. This type of soil allows the implementation of agricultural machinery when managed under periods of adequate humidity. Any natural acidification caused by the high presence of aluminum values can be corrected by commonly used liming methods [43,44].

Figure 2 shows a plane of the elevation profile for the irrigation pivot studied. The boundary of this pivot is outlined in red, and a brown transect represents the landscape’s gradient. The digital model (Figure 2) can also help understand the 30-m elevation variation of the landscape.

### 2.3. Proximal Soil Sensing EM38-MK2

The principle of operation of the EM38-MK2 (Geonics, Mississauga, ON, Canada) is based on the generation of an electromagnetic field through a transmitter coil, which induces eddy currents in the soil. These currents, in turn, create a secondary magnetic field detected by a receiver coil. The strength of this secondary field is directly related to the electrical conductivity of the soil, which can be influenced by various factors such as soil texture, moisture content, salinity, and temperature [45]. The sensor operates in horizontal and vertical dipole modes, allowing for measurements at different depths, typically within a range of 0.75 to 1.5 m, depending on the orientation and configuration of the device. From the study area in Figure 2, the EM38-MK2 sensor was used horizontally, and the coil was used at a 1-m distance for soil volume up to 0.75 cm deep.

Before starting field measurements, the EM38-MK2 is placed on the calibration cane in a horizontal position (Figure 3A), ensuring that the cane supports the sensor at a known and consistent height above the ground. This setup creates a stable and predictable environment where the sensor’s response to the induced electromagnetic field can be assessed without the influence of varying soil conditions. The calibration can ensure that the sensor operates within its expected range and that the output is accurate relative to known standards. The sensor readings are then adjusted to reflect this baseline, compensating for any deviations due to environmental factors such as temperature or instrument drift. Figure 3B shows the sensor configuration for the one-second interval to perform each reading. It was coupled to an all-terrain vehicle driven at a 15 km/h speed.

An exploratory analysis of the original EM38-MK2 data was performed to investigate possible outlier values due to the potential for electromagnetic interference by the metal parts with which the irrigation pivot is constructed. The interferences generated very high electrical conductivity values in specific highly conductive locations. Also, to remove the conductivity reading points made very close to each other collected during brief shutdowns for operational maintenance, the zerodist function present in the sp package of the R software (V. 4.3.3) was applied. Furthermore, the existing electrical conductivity points displaced from the transect format were removed so that the walking simulations with the sensor would get as close as possible to sampling in the format of parallel transects, resulting in a final dataset of 4306 aEC points.

### 2.4. Sampling Designs for the Apparent Electrical Conductivity of EM38-MK2

To characterize the study area, 25 transects were covered with the sensor spaced 40 m apart. This path was treated as an exhaustive design, with greater precision for preparing maps of apparent electrical conductivity, and was, therefore, the best characterized for digital twins (Figure 4).

To simulate a digital twin with little data, reducing sampling costs while maintaining the accuracy of the final electrical conductivity map, a second walk with the EM38-MK2 sensor was configured with only four lines and 400 m spacing between each transect line. This was understood to be the shortest possible distance a producer must travel to produce a transect map (Figure 4).

In addition to the distance between the walking lines being at least twice as large as an Exhaustive Grid for a Sparse Grid, the number of points present in the first is 3906 points while the second and sparce is only 162 reading points (Figure 4).

From the original grid of apparent electrical conductivity, that is, before the data groups for the Exhaustive and Sparse grids were separated, a set of 400 points of apparent electrical conductivity was separated to be used as external validation data for the maps to be produced and compared (Figure 4).

The mapping method considered to be the reference is ordinary kriging using Exhaustive Grid data. This map served as an optimal target during the digital twin stage. The mapping method using the Sparse Grid of the apparent electrical conductivity data was kriging with external drift and geographically weighted regression.

After obtaining an exhaustive and sparse grid, the next step was to optimize the mapping of the sparse grid to generate an electrical conductivity map with an error close to that of the map with an exhaustive grid. Satellite data from a digital elevation model and satellite images were obtained to be fused into the sparse aEC dataset.

### 2.5. Remote Sensing Variables

As the data from proximal sensors were obtained in the first field campaign (September 2018) and the data analyzed in the laboratory were from the second campaign (October 2019), it was decided to select data from remote sensing referring to the two dates of the campaigns to contemplate the two scenarios in which the soil was.

To complement the remote sensing covariate information, the digital elevation model (DEM) obtained by the Alos Palsar satellite with a spatial resolution of 12.5 m was used, and ten relief covariates were derived using the RSAGA package [46] present in the R software [47], and the covariables evaluated were the topographic variables such as aspect, elevation, slope, curvature plan, curvature depth, convergence, topographic wetness index, length-slope factor, relative slope position, channel network distance, and channel network base level.

The Aster satellite bands ast_B1, ast_B2, and ast_B3N were used with wavelength ranges of 0.52–0.60, 0.63–0.69, and 0.78–0.86 μm, respectively.

The data provided by the Sentinel 2 satellite, managed by the European Satellite Agency, included sent_year_B2, sent_year_B3, sent_year_B4, sent_year_B8, sent_year_B5, sent_year_B6, sent_year_B7, sent_year_B8A, sent_year_B11, and sent_year_B12. The data covers diverse wavelengths ranging from 0.44 to 2.31 μm. These variables present optical bands at 10 m resolution and SWIR bands at 20 m resolution.

The Landsat 8 satellite data, organized and distributed by NASA, were used as land_year_B1 to land_year_B11, offering imagery across wavelengths from 0.43 to 12.51 μm at a resolution of 30 m.

Figure 5 provides a comprehensive visual representation of the various covariates used in the study, including digital elevation model (DEM) derivatives and satellite imagery from multiple sensors and time points.

The leftmost column illustrates the DEM derivatives, showcasing topographic features derived from the Alos Palsar satellite with a spatial resolution of 12.5 m. These derivatives include various topographic variables crucial for understanding the landscape’s influence on soil properties.

Moving to the right, the subsequent columns display the spectral images obtained from different satellite platforms. The Aster satellite imagery is shown next, with its bands capturing specific wavelength ranges that contribute to the analysis. Following this, Sentinel 2 satellite images are presented for both the 2018 and 2019 campaigns, highlighting the temporal changes in the spectral characteristics of the study area. Finally, the Landsat 8 satellite images for 2018 and 2019 are displayed, providing additional spectral data across a broad range of wavelengths.

Each set of images demonstrates the variability in the landscape as captured by the different sensors and during the different periods, emphasizing the importance of multi-temporal and multi-sensor approaches in environmental monitoring and modeling. This figure helps to visualize the spatial distribution of the covariates and their changes over time, supporting the methodological choices made in the study.

### 2.6. Preselection of Covariates

The methodology employed for selecting relevant covariates and developing a regression model to predict apparent electrical conductivity (aEC) using the sparse dataset took a stepwise approach, involving the computation of correlations, filtering of covariates, and final model fitting.

#### 2.6.1. Correlation Matrix Computation

A correlation matrix was computed using Pearson correlation for the set of covariates, including 49 raster layers and the target variable (aEC). The correlations were calculated pairwise to handle missing data, ensuring that all available data points were used as described in the following Equation (1):(1)$$cor_{aEC_{sparsed}}=cor\left(X,\;use=pairwise.complete.obs\right)$$
where X represents the matrix of covariates. The diagonal elements of this correlation matrix were set to NA to avoid considering self-correlation (autocorrelation).

#### 2.6.2. Reshaping and Filtering of Correlations

The correlation matrix was then reshaped from a wide format into a long table format, facilitating the identification of the highest correlation values for each variable. The filtering step retained only those pairs of variables with an absolute correlation value greater than or equal to 0.9, as described in Equation (2).
(2)$$\max cor_{aEC_{sparsed}}=\left\{Var_{1},\;Var_{2}:\;\left|cor\left(Var_{1},\;Var_{2}\right)\;\right|\;\geq 0.9\right\}$$

#### 2.6.3. Elimination of Redundant Covariates

For each pair of highly correlated variables identified in the previous step, one variable was dropped based on its lower correlation with the target variable aEC. Specifically, for each correlated pair ($Var_{1},\;Var_{2}$), the variable with the smaller correlation to aEC was eliminated using Equation (3):(3)$$drop\;variables_{aEC_{sparsed}}=\min\left(cor\left(aEC,\;Var_{1}\right),\;cor\left(aEC,\;Var_{2}\right)\right)$$

The remaining covariates formed the final set of independent variables for subsequent modeling.

#### 2.6.4. Subset Selection Using Exhaustive Search

To identify the most relevant subset of covariates, an exhaustive subset selection method was employed using the regsubsets function, optimizing for the Bayesian information criterion (BIC) and adjusted R-squared (*R*^2^) using Equation (4):(4)$$RegSubset_{aEC_{sparsed}}=\arg min_{subset}\;BIC\left(subset\right)\;\mathrm{and}\;\arg max_{subset}\;adj\;R^{2}\left(subset\right)$$

The optimal subset of covariates was determined based on the smallest BIC and the highest adjusted *R*^2^.

### 2.7. Mapping Methods

#### 2.7.1. Ordinary Kriging—Reference Map

The original dataset contained 3906 apparent electrical conductivity (aEC) points and was used as the reference grid in the digital twin approach. To achieve a normal distribution of the aEC data, the aEC for the Neperian log was transformed.

A semivariogram was adjusted for this set of points. The data were spatialized by ordinary kriging (OK) using the krige function of the gstat package [48] from the R software.

The semivariogram estimates a value in a region with a known distance using data near the estimation site [49]. In this way, ordinary kriging uses just the distance between points to comprehend the phenomena of the distribution of the aEC, for instance.

The aEC sparse data was also mapped to show how a mapping via OK of the sparse data would be. All maps were interpolated using a spatial resolution of 10 m.

#### 2.7.2. Kriging with External Drift

The external drift kriging (KED) method was used to map the sparse aEC data associated with the remote sensing trend. KED incorporates the local trend within a neighborhood search window as a linear function of a mildly varying secondary variable, and the trend of the primary variable must be linearly related to the secondary variable [50].

Therefore, KED interpolated the apparent electrical conductivity using the gstat package’s krige function [48] associated with preselected satellite covariables. The KED aEC map’s final resolution was 10 m, compared to the aEC reference map using ordinary kriging.

#### 2.7.3. Geographically Weighted Regression

Geographically weighted regression (GWR) was also applied to the sparse aEC dataset. GWR accounts for spatial non-stationarity by allowing local rather than global parameter estimation. The regression model was constructed with selected remote sensing and relief covariates as predictors and the aEC as the response variable. The local coefficients were estimated using the spgwr package [51] in R, and the predicted aEC values were mapped across the study area. The GWR aEC map’s final resolution was 10 m, compared to the aEC reference map using ordinary kriging.

### 2.8. Maps of Irrigation Management Zones and Efficiency Assessment

To associate apparent electrical conductivity (aEC) with soil moisture levels, we categorized the aEC values into three classes based on the ordinary kriging map generated from the exhaustive aEC dataset. The range of aEC values less than 8.44 mS/m was designated as “Dry”, the range from 8.44 to 13.77 mS/m was classified as “Intermediate”, and the range from 13.77 to 19.11 mS/m was classified as “Moist”. These categories were established to provide a practical framework for interpreting the aEC data in terms of moisture content within the soil, facilitating the identification of areas with varying moisture levels.

Four management zone (MZ) maps for irrigation were defined using the aEC exhausted map via OK, the aEC OK map using the sparse dataset, the aEC map using the KED method, and the aEC map using the GWR. The kmeans function was used in the stats package natively present in the R software. The four electrical conductivity maps were organized in a data matrix format and grouped using the *k*-means method. This method partitioned the dataset into *k* groups.

In the present study, three zones were chosen as the *k* value, as many would not be operationally practical for farmers. All the data used in the kmeans function were parameterized to the values of zero, mean, and variance one, using the scale function present in the stats package of the R software.

Apparent electrical conductivity (aEC) is a valuable indirect measurement for assessing soil moisture and clay content due to how these properties influence the soil’s ability to conduct electricity. Moisture content significantly impacts aEC, as water enhances the soil’s conductivity, leading to higher aEC values in wetter soils. Similarly, clay-rich soils exhibit higher aEC because clay particles have a large surface area and a high capacity to retain water and ions, contributing to increased conductivity. The mineralogical composition of clay also enhances electrical conductivity, making aEC a common proxy for both moisture and clay content. Given the strong relationship between clay content and soil moisture retention, the agreement between zones identified by aEC and those supported by clay texture data provides some confidence that these zones may reflect variations in soil moisture, even if direct moisture measurements are unavailable.

To evaluate the potential of using a sparse aEC dataset to produce a map similar to the exhaustive dataset, as well as the effectiveness of these maps when used as input information in the *kmeans* process to define management zones for irrigation, data from 72 soil sampling locations at 0–10 cm depth were collected (Figure 6) and analyzed for clay content using the sieve and pipette method.

The 72 points analyzed in the laboratory validated the four MZ maps produced from aEC mapped via OK, KED, and GWR. The zone classes were extracted from the two MZ maps for the 72 coordinates associated with the laboratory data using the extract function present in the raster package in the R software.

Finally, the analysis of variance (ANOVA) was used to identify whether the management zones could distinguish the variance of the values of the laboratory attributes regarding their classes of management zones. The aov function of the stats package in the R software was used for that.

### 2.9. Map Accuracy

From the filtered electrical conductivity dataset containing 26 lines, 400 points were selected randomly using the sample function in R. From this subset of 400 points, metrics were calculated to evaluate the accuracy of the generated aEC maps. The accuracy values of the maps generated by KED, GWR, and OK using the sparse aEC dataset were compared to the aEC reference map produced by OK from the exhaustive dataset.

The metrics used to compare the maps were mean error (ME—Equation (5)), mean absolute error (MAE—Equation (6)), square root of the mean square error (RMSE—Equation (7)), the relationship between performance and deviation (RPD—Equation (8)), and the coefficient of determination (R^2^—Equation (9)), as follows:(5)$$ME=\frac{1}{n}\overset{n}{\underset{i=1}{\sum }}{\left(y_{i}-{ŷ}_{i}\right)}^{2}$$
(6)$$MAE=\frac{1}{n}\overset{n}{\underset{i=1}{\sum }}\left|y_{i}-{ŷ}_{i}\right|$$
(7)$$RMSE=\sqrt{\frac{1}{n}\overset{n}{\underset{i=1}{\sum }}{\left(y_{i}-{ŷ}_{i}\right)}^{2}}$$
(8)$$RPD=\sqrt{\frac{Standard\;Deviations}{RMSE}}$$
(9)$$R^{2}=1-\frac{\sum_{i=1}^{n}{\left(y_{i}-{ŷ}_{i}\right)}^{2}}{\sum_{i=1}^{n}{\left(y_{i}-{\bar{y}}_{i}\right)}^{2}}$$
where *n* is the number of observations; $y_{i}$ are the actual values; ${ŷ}_{i}$ are the predicted values; ${\bar{y}}_{i}$ is the average of the actual values of $y_{i}$.

## 3. Results

### 3.1. Exploratory Data Analysis

The descriptive statistics for the aEC exhausted, and the original dataset is described in Table 1. For the training set with 25 rows of logarithmic scale data, the apparent electrical conductivity (aEC) ranged from 0.96 to 3.27 millisiemens per meter (mS/m), with a first quartile (25th percentile) of 1.92 and a third quartile (75th percentile) of 2.45. The mean was approximately 2.2, and the median was 2.23. The variance was 0.13, and the standard deviation was 0.36, indicating a relatively concentrated distribution around the mean. The skewness was −0.09, suggesting a slight tail to the left in the distribution, while the kurtosis was −0.58, indicating a relatively flattened distribution compared to a normal distribution.

For the external aEC validation dataset, the aEC ranges from 2.62 to 26.25, with a first quartile of 6.84 and a third quartile of 11.56. The mean is approximately 9.58, and the median is 9.30. The variance is 11.49, and the standard deviation is 3.39, indicating a wider data dispersion than the training set. The asymmetry is 0.68, indicating a tail to the right in the distribution, while the kurtosis is 0.22, which is slightly closer to normal distribution than the training set.

For the sparse aEC dataset, the aEC ranged from 3.87 to 18.67, with a first quartile of 7.40 and a third quartile of 11.48. The mean is approximately 9.94, and the median is 10.39. The variance is 9.67, and the standard deviation is 3.11, indicating a moderate dispersion of the data. The skewness is 0.28, suggesting a slight tail to the right, while the kurtosis is −0.22, indicating a slightly less concentrated distribution in the tails than a normal distribution.

The electrical conductivity data in their original format showed a slight grouping on the left, i.e., with a tail on the right (Figure 7A). Thus, Neperian logarithm transformation was used to normalize the data and use ordinary kriging (Figure 7B).

### 3.2. Predictive Model

Based on the pre-processing presented in Section 2.2, the best model for predicting aEC using the sparse dataset is shown in Table 2. The “landforms_tpi_based” variable has a negative coefficient of −0.18. This implies that areas associated with this attribute show reduced aEC. In other words, as the “landforms_tpi_based” values increase, the aEC decreases.

The variable “curv_total” shows a considerable influence, indicated by the high coefficient value of −9424.44. This suggests that even a slight increase in “curv_total” is associated with a significant drop in aEC value.

Analyzing other variables such as the satellite bands “land_2018_B5”, “land_2018_B10”, “land_2019_B2”, “land_2019_B6”, “sent_2018_B8A”, and “sent_2019_B3”, all have negative coefficients close to zero. This means that even subtle changes in these attributes are linked to decreased aEC, although this relationship is relatively tiny. This is interesting because groups of satellite bands from different years were selected from 2018 to 2019.

It is important to note that all the variables mentioned are statistically significant for predicting electrical conductivity, as indicated by the small *p*-values. The coefficient of determination (R^2^) shows that the independent variables in the model can explain approximately 87.2% of the variation of aEC. For example, 1–0.872 indicates that approximately 13% of the variability of the aEC distribution phenomenon “remained” to be modeled in the semivariogram adjustment stage. Figure 8 shows the adjustment line of the linear model adjusted for the aEC sparse data as a function of the remote sensing data.

### 3.3. Semivariograms

The aEC semivariogram for an exhaustive dataset using OK was adjusted by the spherical model, and a minimal random variability (nugget) and significant spatial correlation at distances up to 500 m (range) were observed (Figure 9A). This semivariogram is treated as a reference for comparison with the OK, KED, and GWR mapping methods when using the aEC sparse dataset.

The semivariograms using the sparse aEC dataset and OK were better adjusted with an exponential model, and it revealed a slight increase in random variability (nugget) and a significant spatial correlation at shorter distances of 250 m (Figure 9B). In the case of KED, the random variability is more substantial (more considerable nugget value), and the spatial correlation is strong only over a 25-m range (Figure 9C).

The KED mapping method shows that the 13% variability still showed spatial dependence, even at a range of 25 m. Thus, the semivariogram implemented by the KED model captured the entire spatial dependency structure of aEC variability in the sparse dataset.

The application of GWR using the sparse dataset showed minimal random variability (nugget) and a robust spatial correlation at distances up to 233 m (Figure 9D). In contrast, the spatialization of the R^2^ values obtained by the GWR model from a semivariogram adjusted for the Gaussian model shows very low structured variability and a significant spatial correlation limited to 130 m (Figure 9E). Compared to KED, the GWR method presented almost 10× more value of spatial dependence distance (range) despite using the same covariates for point-to-point prediction. These range values cannot be comparable because, for KED, we are talking about the residuals of the regression model, while in the case of GWR, we are talking about spatialization of the predicted aEC values via the GWR model and subsequent spatialization by OK.

### 3.4. Electrical Conductivity Maps and External Validation

#### 3.4.1. Maps

The aEC maps using the exhaustive dataset are shown in Figure 10A. The aEC values were standardized for equal intervals as follows: <8.44; <8.44 and <13.77; <13.77 and <19.11; and >19.11 millisiemens per meter (mS/m). In this way, we can compare the visual patterns produced by the different mapping methods. Still, in the aEC reference map, red represents the lowest aEC values, while blue represents the highest values. The green area in Figure 10A shows the high aEC values located where there is a natural drainage.

The aEC OK map using the sparse dataset shows a non-representative distribution of the aEC values, as it is possible to compare to the aEC reference map. The absence of aEC information between transects significantly impacted the production of the aEC map when the mapping procedure was OK.

However, when the sparse aEC dataset is combined with the KED mapping method, the spatial pattern of the aEC (Figure 10C) is like the aEC reference map (Figure 10A). It represents the success of the data fusion between proximal and remote datasets and positively proves the present hypothesis.

Despite the well-adjusted model fitted between the aEC sparse dataset and the satellite covariables, the GWR method for mapping did not present a benefit for use, and the aEC map presented in Figure 10D does not resemble the aEC reference map.

The point-to-point R^2^ fit values for the aEC mapping via GWR can be visualized by looking at the fit map of the data in Figure 10E. The smallest fit values (<0.78) are close to the original position of the aEC sparse dataset coordinates. In this sense, the relationships between the dependent and independent variables may have been compromised since the covariates may not have had good correlations at these distances.

#### 3.4.2. External Validation

The aEC map using OK and the exhaustive dataset presented the coefficient of determination (R^2^) reaching 97. In addition, the mean error (ME) is zero, meaning the model has no systematic bias. The mean absolute error (MAE) is low, indicating an average accuracy of 0.45 in the predictions. The mean square error (RMSE) is also low, with a value of 0.56, indicating a precision in the predictions. The performance deviation ratio (RPD) is relatively high, reaching 5.86, which suggests the model’s excellent ability to predict data variability. The summarized external validation results are shown in Table 3.

In contrast, the aEC uses OK, but considering the sparse dataset, although the R^2^ is still relatively high (0.71), the other metrics show lower performance than the reference aEC map. The MAE increases to 1.41 and the RMSE to 1.87, indicating a reduced prediction accuracy, while the RPD decreases to 1.25, suggesting a less reliable forecasting ability concerning data variability.

The results of the KED mapping for the aEC for the sparse dataset are slightly better than the aEC and OK method for the sparse dataset but are still inferior to the aEC OK map for the exhaustive dataset. The R^2^ is 0.78, indicating a relatively good fit to the aEC data, while the MAE is 1.26 and the RMSE is 1.62, suggesting moderate accuracy in the forecasts. The RPD is 1.76, indicating reasonable predictability concerning data variability. We can see that the KDE method is better than just using the target variable’s spatial dependence compared to the aEC mapping using OK and sparse datasets.

GWR mapping method for aEC and sparse dataset are the weakest among the methods evaluated. The R^2^ is 0.57, indicating a modest fit to the data, while the MAE is 1.78 and the RMSE is 2.30, indicating relatively low prediction accuracy. In addition, the RPD is 0.81, indicating limited predictive ability concerning data variability. The GWR method performed worse than the aEC OK mapping procedure when using the sparse aEC dataset.

### 3.5. Management Zones and Zone Validation

#### 3.5.1. Management Zones Maps

The irrigation MZ maps are exhibited in Figure 11. The OK map of the aEC using the exhaustive dataset presented the irrigation MZ reference map (Figure 11A). The labels were created to associate the zones in the maps with our inferences regarding the capacity to retain the water in the pivot area and consider the values of the aEC and the moisture’s behavior [52,53]. This way, blue is Dry, yellow is Intermediate, and green is Moist.

As the k-means process just handled one piece of information, the output resembles the input information, as in the case of the irrigation MZ map made using the aEC sparse dataset and OK, producing different irrigation borders in Figure 11B. In the same way, the GWR aEC map produced a different pattern from the reference irrigation MZ map. On the other hand, the irrigation MZ map using KED and aEC sparse data (Figure 11C) produced the same MZ pattern as the reference MZ.

From the MZ map (Figure 11A) created from the aEC map taken as the reference map or the one obtained by the KED method (Figure 9C), we could return to the producer and recommend that they irrigate the area presented in green for less time than the yellow and blue areas, which should have more irrigation time.

#### 3.5.2. Validation of Management Zones

The management zoning process using the OK aEC exhaustive dataset shows in Table 4 the ANOVA for the reference irrigation MZ map when the efficiency was tested to distinguish the irrigation zones when comparing the texture properties.

For the irrigation MZ map using the aEC sparse dataset and the OK method, although the Scott–Knott test indicated a possible significant difference, the ANOVA did not confirm this discrepancy (Pr > Fc = 0.07). Therefore, there is insufficient statistical evidence to state that the irrigation management zones map in this combination should present any benefit to treating the digital twin heterogeneity of the soil.

The irrigation MZ map using the aEC sparse dataset and GWR method showed significant differences in the reference treatment (Pr > Fc = 0.03). The Scott–Knott test also corroborated these differences, providing statistical support (Table 4).

The irrigation MZ map using the KED aEC sparse dataset and GWR presented a similar potential to distinguish the irrigation boundaries considering the ANOVA. Irrigation MZ maps using the sparse aEC dataset and KED and GWR mapping methods demonstrated statistically significant differences from the reference procedures. Hence, the final decision should consider additional factors such as cost, practicality of implementation, or other criteria relevant to the study context.

For example, although both were efficient in distinguishing treatment areas, we can understand that operationally, the management zone map produced by the KED aEC sparse dataset presents spatial patterns compatible with the possible alteration of an irrigation pivot commonly found in Brazil. It is possible to set the irrigation pivot speed up or down in certain areas, considering a “pie slices” format. On the other hand, considering the irrigation MZ map by GWR, it could be challenging to recommend speeding up the rotation since the zone patterns are in the “amorphous” format.

## 4. Conclusions

The incorporation of remote sensing data significantly enhanced the accuracy of mapping apparent electrical conductivity (aEC), even when a sparse dataset was used. Specifically, when remote sensing data was combined with the kriging with external drift (KED) method, the resulting aEC map achieved a relatively high R^2^ value of 0.78, outperforming both the ordinary kriging (OK) method applied to the sparse dataset (R^2^ = 0.71) and the geographically weighted regression (GWR) method, which had the weakest performance with an R^2^ of 0.57. The KED method also demonstrated moderate prediction accuracy with a mean absolute error (MAE) of 1.26 and a root mean square error (RMSE) of 1.62, indicating its effectiveness in mapping aEC in scenarios with limited data.

When evaluating irrigation management zones (MZs), the map generated using the exhaustive aEC dataset and the OK method was able to delineate three distinct zones for differentiated irrigation treatment. These zones were validated using soil texture data, confirming the map’s ability to identify areas with higher soil moisture conservation. Furthermore, the irrigation MZ map produced using the sparse dataset and the KED method could distinguish irrigation zones with comparable precision to the reference MZ map. The ANOVA results for the MZs identified by the KED method were consistent with those from the reference map, underscoring the reliability of remote sensing data and KED in precision agriculture, even with reduced sampling density.

## Figures and Tables

**Figure 1 sensors-24-05742-f001:**
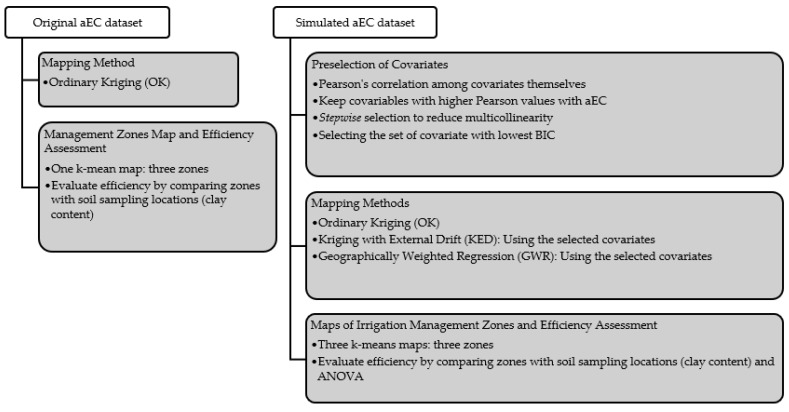
Flowchart of the methodology of simulation of the aEC dataset with sparse sampling and the mapping methods followed by the management zones approach.

**Figure 2 sensors-24-05742-f002:**
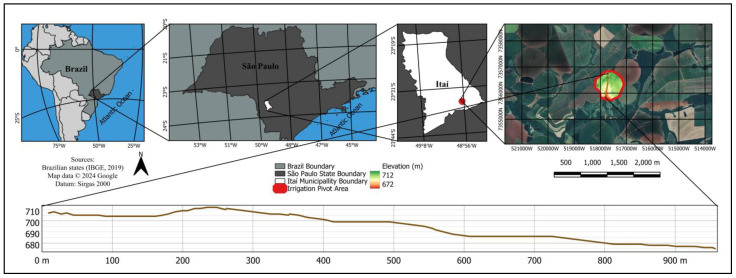
Location of the study area with the height gradient and digital elevation model.

**Figure 3 sensors-24-05742-f003:**
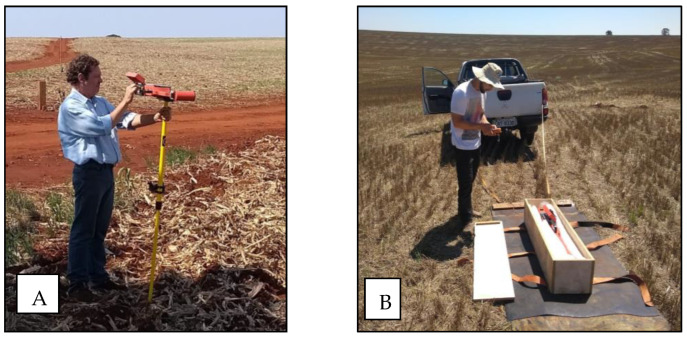
(**A**) EM38-MK2 being calibrated to the specific magnetic scenario of the field; (**B**) the sensor is paired with the handheld controller to set the timing acquisition.

**Figure 4 sensors-24-05742-f004:**
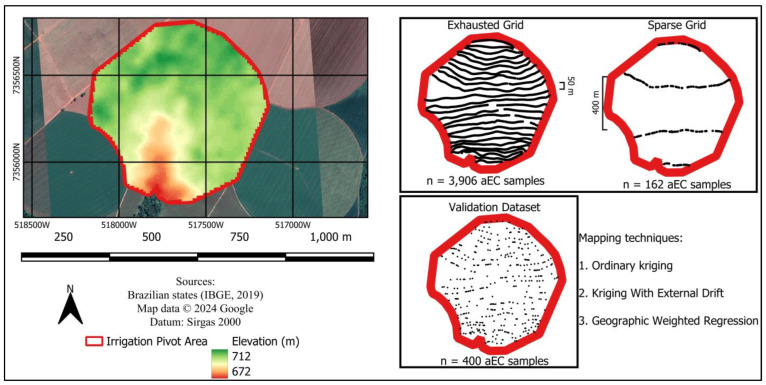
Study area with digital elevation model in the background; Exhaustive Grid and Sparse Grid showing the distance between sampling lines; external validation dataset.

**Figure 5 sensors-24-05742-f005:**
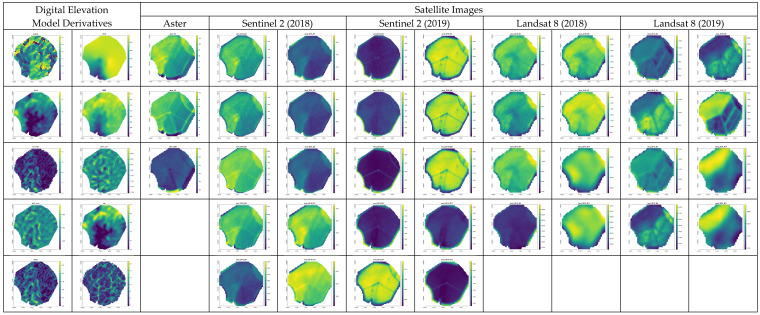
Remote sensing covariates used as predictors in the kriging with external drift and geographically weighted regression.

**Figure 6 sensors-24-05742-f006:**
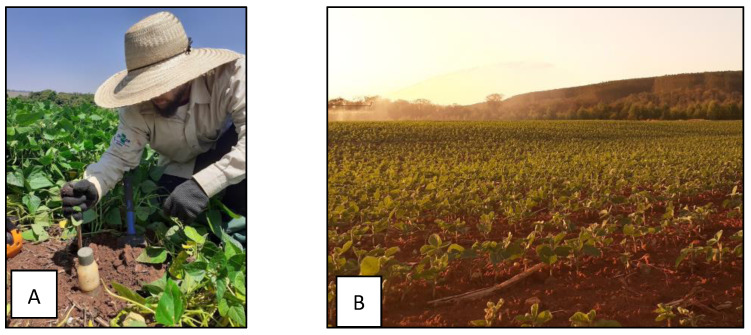
(**A**) Soil sampling of 0–10 cm using soil sampler ring; (**B**) planting area covered by beans and irrigated by a central pivot on the background.

**Figure 7 sensors-24-05742-f007:**
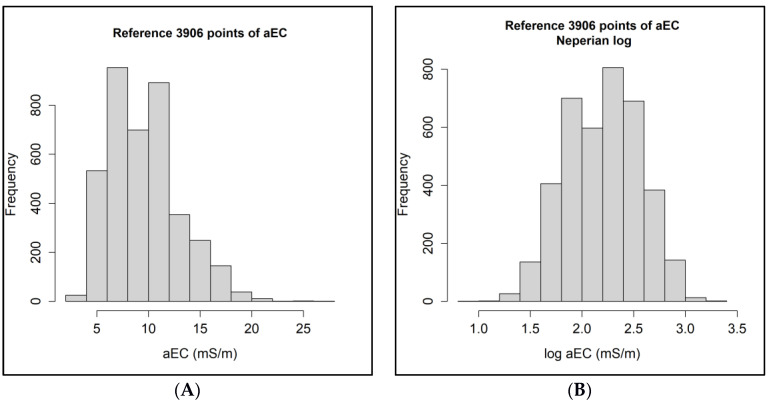
Electrical conductivity data in mS/m (millisiemens per meter); (**A**) original format; (**B**) transformed to Neperian logarithm.

**Figure 8 sensors-24-05742-f008:**
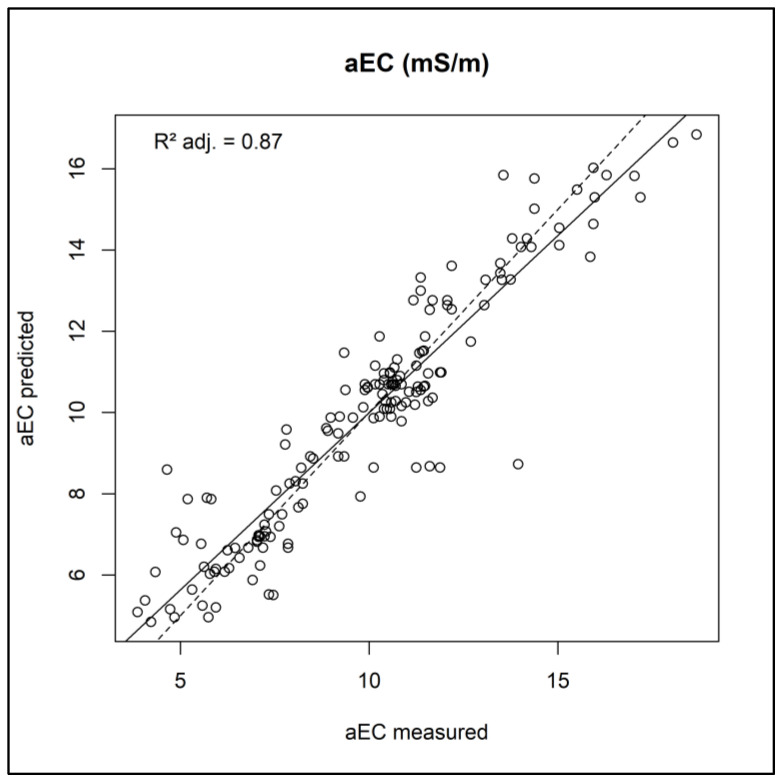
Predicted versus experimental aEC values of proximal sensor data as a function of remote sensor data using the training dataset. The continuous black lines adjust the intercept and slope for the models, while the dashed lines are intercepted and idealized as 1 and 0, respectively. R^2^ adj: R^2^ adjusted value; aEC: apparent electrical conductivity in mS/m (millisiemens per meter).

**Figure 9 sensors-24-05742-f009:**
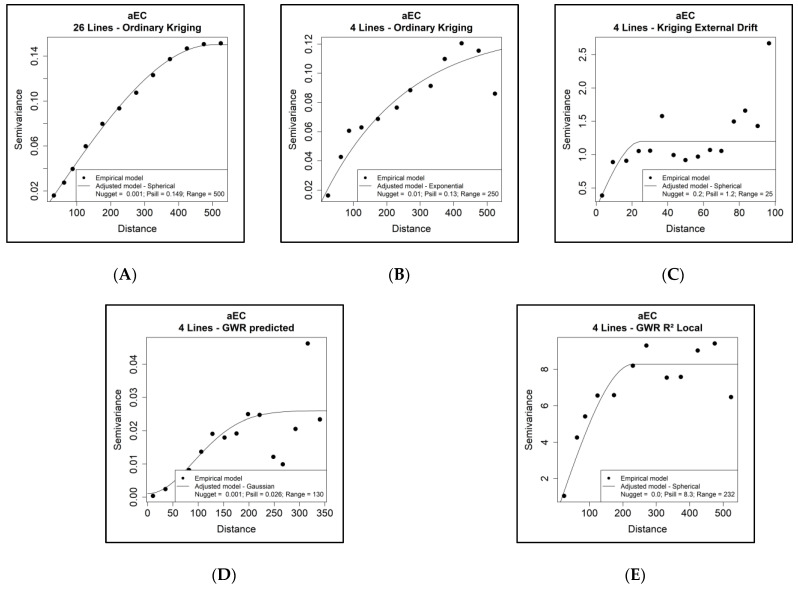
Empirical (circles) and adjusted (lines) semivariograms of apparent electrical conductivity (aEC) in mS/m. (**A**) Using ordinary kriging with 26 lines (reference); (**B**) using ordinary kriging with four rows (sparse); (**C**) using kriging with external drift of aEC data with sparse data as a function of remote sensor data defined in Section 3.5; (**D**) using geographically weighted regression with sparse data as a function of remote sensor data defined in Section 3.5; (**E**) semivariogram of the R^2^ indices obtained by calculating the GWR for spatialization.

**Figure 10 sensors-24-05742-f010:**
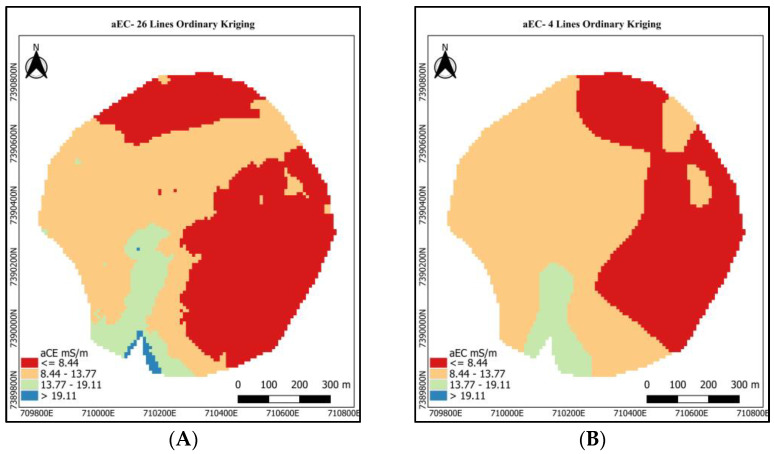
Maps of apparent electrical conductivity (aEC) in mS/m. (**A**) Using ordinary kriging with 26 lines (reference); (**B**) using ordinary kriging with four rows (sparse); (**C**) using kriging with external drift of the sparse data as a function of the remote sensing data defined in Section 3.5; (**D**) using geographically weighted regression with sparse data as a function of remote sensor data defined in Section 3.5; (**E**) map of the adjusted R^2^ obtained by calculating the GWR for the aEC sparse dataset.

**Figure 11 sensors-24-05742-f011:**
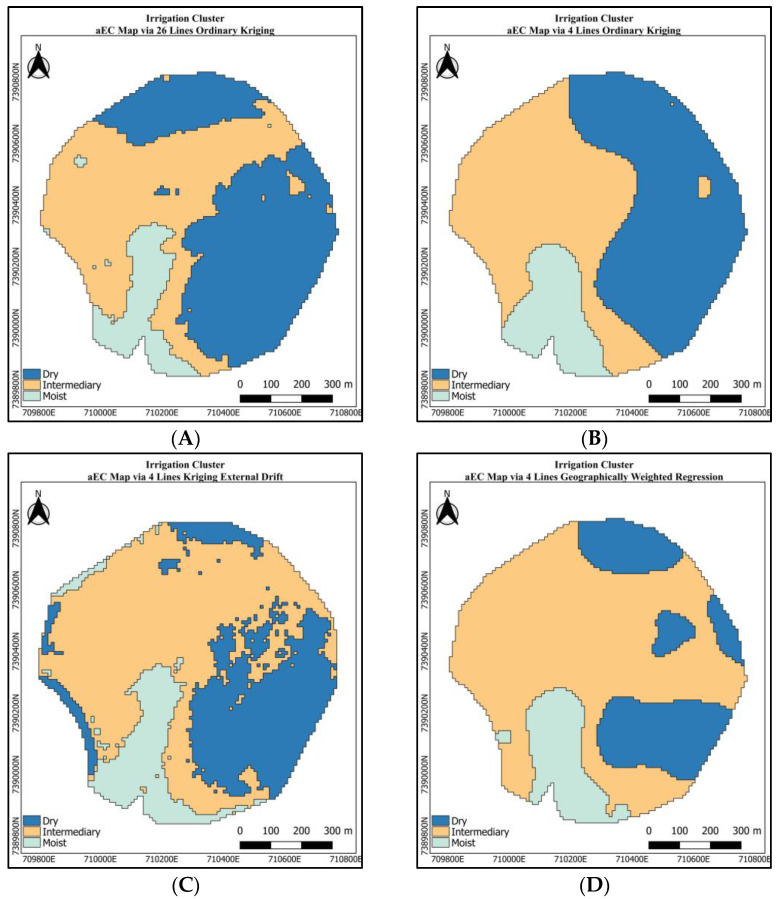
Maps of management zones for soil types. (**A**) Using ordinary kriging with 26 lines (reference); (**B**) using ordinary kriging with four rows (sparse); (**C**) using kriging with external drift of the sparse aEC data as a function of the remote sensing data defined in Section 2.6.2; (**D**) using geographically weighted regression with sparse data as a function of remote sensor data defined in Section 2.6.3.

**Table 1 sensors-24-05742-t001:** Descriptive statistics of electrical conductivity data mS/m.

	aEC Exhausted (log)	aEC Exhausted	aEC Sparse	aEC External Validation
n	3906	3906	162	400
Minimum	0.96	2.62	3.87	3.63
Maximum	3.27	26.25	18.67	25.31
1. Quantile	1.92	6.84	7.40	6.87
3. Quantile	2.45	11.56	11.48	11.49
Mean	2.20	9.58	9.94	9.62
Median	2.23	9.30	10.39	9.53
Variance	0.13	11.49	9.67	11.2
Standard Deviation	0.36	3.39	3.11	3.35
Skewness	−0.09	0.68	0.28	0.78
Kurtosis	−0.58	0.22	−0.22	0.92

**Table 2 sensors-24-05742-t002:** Adjustment parameters of the linear electrical conductivity model as a function of remote sensing data using the best combination of Pearson’s selection, in addition to the BIC criterion and, finally, using a regsubset function.

aEC (mS/m)			
Coefficient	Estimated	Confidence Interval (95%)	*p*-Value
(Intercept)	29.20	1.34–57.05	<0.05 *
landforms_tpi_based	−0.18	−0.31–−0.06	<0.05 *
curv_total	−9424.44	−14,713.09–−4135.80	<0.05 *
land_2018_B5	−0.00	0.00–0.00	<0.05 *
land_2018_B10	−0.00	0.00–0.00	<0.05 *
land_2019_B2	0.01	0.01–0.01	<0.05 *
land_2019_B6	−0.00	0.00–0.00	<0.05 *
sent_2018_B8A	0.01	0.01–0.02	<0.05 *
sent_2019_B3	−0.04	0.05–−0.03	<0.05 *
Observations	162		
R^2^/R^2^ adjusted	0.872/0.866		

* Significant at a level of 5%.

**Table 3 sensors-24-05742-t003:** Accuracy values of the maps obtained via external validation using the 400 aEC points.

R^2^	ME	MAE	RMSE	RPD
OK—Exhaustive Dataset				
0.97	0.00	0.45	0.56	5.86
OK—Sparse Dataset				
0.71	0.00	1.41	1.87	1.25
KED—Sparse Dataset				
0.78	0.00	1.26	1.62	1.76
GWR—Sparse Dataset				
0.57	0.00	1.78	2.30	0.81

**Table 4 sensors-24-05742-t004:** Analysis of variance (ANOVA) of MZ classes created from aEC maps via OK 26 and 4 lines, via KED and GWR via 4 aEC lines.

aEC 26 Lines OK											
	Treatment	Scott–Knott	Mean	GL	SQ	QM	Fc	Pr > Fc	CV (%)	Shapiro–Wilk (*p*-value)	Homogeneity of Variances (*p*-value)
Treatment	1	a—Moist	431.11	2	20,067.00	10,033.40	5.51	0.01	10.32	0.00	0.00
Residue	2	a—Dry	425.88	69	125,533.00	1819.30					
Total	3	b—Intermediate	393.10	71	145,600.00						
aEC 4 Lines OK											
	Treatment	Scott–Knott	Mean	GL	SQ	QM	Fc	Pr > Fc	CV (%)	Shapiro–Wilk (*p*-value)	Homogeneity of Variances (*p*-value)
Treatment	1	a	401.33	2	10,806.00	5402.90	2.77	0.07	10.69	0.00	0.00
Residue	2	a	438.00	69	134,794.00	1953.50					
Total	3	a	416.87	71	145,600.00						
aEC 4 Lines KED											
	Treatment	Scott–Knott	Mean	GL	SQ	QM	Fc	Pr > Fc	CV (%)	Shapiro–Wilk (*p*-value)	Homogeneity of Variances (*p*-value)
Treatment	1	a—Moist	431.11	2	13,760.00	6,879.90	3.60	0.03	10.58	0.00	0.00
Residue	2	a—Dry	427.82	69	131,840.00	1910.70					
Total	3	b—Intermediate	401.00	71	145,600.00						
aEC 4 Lines GWR											
	Treatment	Scott–Knott	Mean	GL	SQ	QM	Fc	Pr > Fc	CV (%)	Shapiro–Wilk (*p*-value)	Homogeneity of Variances (*p*-value)
Treatment	1	a—Moist	440.00	2	14,686.00	7343.10	3.87	0.03	10.54	0.00	0.00
Residue	2	a—Dry	431.25	69	130,914.00	1897.30					
Total	3	b—Intermediate	403.67	71	145,600.00						

## Data Availability

The data disclosed in this research can be obtained upon request from the corresponding author. The non-public availability of the data is attributed to intellectual property considerations by the funding research company.

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
