# Peer review of "Remote and Proximal Sensors Data Fusion: Digital Twins in Irrigation Management Zoning"

_sensors, 2024, doi:10.3390/s24175742_

Round 1

Reviewer 1 Report

Comments and Suggestions for Authors

The authors conducted a study to identify the most effective method for creating irrigation management zones. They tested and validated interpolation and prediction methods using data from sensors and satellites. Overall, the paper is well-structured; however, before publication, I recommend a more thorough revision. Below are the main areas that need improvement:

Data collection at the edges of fields can present significant issues, as these areas may be subject to external influences such as proximity to roads, variations in moisture due to adjacent vegetation, and other environmental factors. These influences can affect the accuracy of irrigation needs estimation. Therefore, it is crucial that the simulation of data collection, especially in regions with lower sample density, is carefully planned, taking these additional variables into account. Proper planning will help minimize the impact of these external influences and improve the accuracy of irrigation estimates.

The methodology should be enhanced for a better understanding and application of the methods used. It is suggested to include a detailed flowchart describing the modeling and methodological processes employed. This will facilitate understanding and transparency of the adopted approach.

Topic 2.1: It is recommended to add a figure or table illustrating the spectral bands and wavelengths associated with each satellite. This will provide a clearer understanding of the spectral characteristics used in data analysis and modeling.

Topic 2.2: The section should be revised to clarify the methodology for variable selection. It is necessary to detail the equations and metrics used to evaluate and select variables. Additionally, it should specify how different variable combinations were tested, including the evaluation metrics used to measure error and accuracy. This information is essential to visualize the robustness of the models and the precision of the predictions made.

A crucial aspect to consider before publication is the methodology used. Although the authors focused on interpolation methods, the validation of these models needs to be improved. They collected 72 soil samples, and the validation of the interpolation methods should concentrate on predicting the amount of sand present in these samples. It is essential to identify which of the methods used provides the best prediction of soil sand content in order to create more precise and efficient irrigation zones.

Comments on the Quality of English Language

Reviewer 2 Report

Comments and Suggestions for Authors

This is an interesting study that focuses on optimizing precision agriculture through soil mapping techniques and contributes to understanding and applying these techniques in the agricultural sector. The integration of remote sensing data with data from proximal sensors is particularly important.
This study aims to optimize resource use, but the initial cost and resources needed for acquiring and processing high-resolution remote sensing data, along with purchasing and operating advanced sensors like the EM38-MK2, could be too expensive for smaller farms or those with limited budgets.

The sparse dataset, with only 162 data points, might not be sufficient to capture the full variability of soil properties in the study area. This could limit the generalizability and reliability of the findings. Why was this number of data points chosen?
Have you tried using different numbers of data points?

The study is conducted in a specific soil type prevalent in São Paulo, Brazil. The applicability and effectiveness of the methods for different soil types, with varying physical and chemical properties, need further exploration.
Please consider this for your future research.

The figures and the captions need to be adjusted to fit the template.

I suggest including more results with numbers to show how much better one method is compared to another, to enhance understanding of the readers and effectiveness of the different methods in the abstract and conclusions.

Reviewer 3 Report

Comments and Suggestions for Authors

The authors conducted a series of studies and processed the obtained data. In justifying their implementation, they often use the concept of precision agriculture and resource efficiency (including the economic acquisition of data on the crop environment). However, they do not indicate the exact, practical purpose of the studies (vehicle routes cannot be considered such a purpose). Using electromagnetic waves to measure the electrical conductivity of soil is not a new technique. However, there are different devices for this type of measurement: precise ones, such as TDR, or area (more geotechnical) ones, such as the meter used by the authors. In the work, the authors modestly describe the soil conditions (including their variability) in the analyzed area. It should be remembered that the value of soil's electromagnetically measured electrical conductivity is closely related to its current moisture (the authors do not write much on this subject). When presenting the research methodology, the authors should present the principle of operation of the measuring device used in them (EM38-MK2) and describe its calibration process and what it depends on. This type of device reacts strongly to interference from underground infrastructure elements (e.g., pipelines supplying pivots). The authors do not describe the impact of interference on the obtained results in detail. The article makes it difficult to understand the purpose of defining irrigation zones and how they can be practically used to control irrigation. The terms wet, average, etc., used by the authors to describe maps are not very precise and of little value from a practical point of view. The research should undoubtedly be supplemented by more accurate validation of the obtained results based on other techniques for measuring electrical conductivity (TDR or other) or soil moisture (e.g., the simplest gravimetric).

Round 2

Reviewer 1 Report

Comments and Suggestions for Authors

Comments on the Quality of English Language